# Artificial Reproductive Technology (ART) Applied to Female Cervids Adapted from Domestic Ruminants

**DOI:** 10.3390/ani11102933

**Published:** 2021-10-11

**Authors:** Anna J. Korzekwa, Angelika M. Kotlarczyk

**Affiliations:** Department of Biodiversity Protection, Institute of Animal, Reproduction and Food Research of Polish Academy of Sciences (IAR&FR PAS), Tuwima 10 Str., 10-748 Olsztyn, Poland; a.kotlarczyk@pan.olsztyn.pl

**Keywords:** cervid, reproduction, fertilization, embryo, cryopreservation

## Abstract

**Simple Summary:**

The application of ART in Cervidae is crucial for conservation, especially those endangered by extinction, management of farmed deer, and understanding cellular mechanisms in the reproductive processes. The reproductive techniques used in domestic ruminants are adapted in female deer. Nowadays, ART includes synchronization of the estrous cycle, artificial insemination, superovulation, oocyte collection, IVM, IVF, embryo transfer, and cryopreservation of oocytes and embryos. Some reproductive biotechniques, such as IVF, have been already adopted for female cervids with satisfactory results, while others, such as cryopreservation of oocytes and embryos, still require refinement. Some environmental factors influence the success of ART, e.g., stress susceptibility.

**Abstract:**

There are about 150 Cervidae species on the IUCN Red List of Threatened Species. Only a small part is counted among farm animals, and most of them are free roaming. The universality and large numbers of representatives of cervids such as red deer (Cervus elaphus) and roe deer (Capreolus capreolus) may predispose these species to be used as models for research on reintroduction or assisted reproduction of deer at risk of extinction. We outlined the historical fluctuation of cervids in Europe and the process of domestication, which led to breeding management. Consequently, the reproductive techniques used in domestic ruminants were adapted for use in female deer which we reviewed based on our results and other available results. We focused on stress susceptibility in cervids depending on habitat and antropopression and proposed copeptin as a novel diagnostic parameter suitable for stress determination. Some reproductive biotechniques have been adopted for female cervids with satisfactory results, e.g., in vitro fertilization, while others still require methodological refinement, e.g., cryopreservation of oocytes and embryos.

## 1. Historical Fluctuations in the Cervidae Population in Europe 

The biodiversity evolution in nature is driven by environmental and biotic factors [1]. According to Ludt et al. [2], at the Oligocene–Miocene boundary, the first cervoids appeared, and Central Asia seems to be the origin of the genus Cervus. Studies have shown the probability of the existence of two different species of red deer, including three subspecies in Asia and America (Eastern Red Deer) and four Eurasian subspecies (Western Red Deer), with an additional one or two primordial subspecies in Central Asia [2,3]. In Europe, a key evolutionary factor is the geography and history of climate change through the Pleistocene. The cyclic climatic changes during the Quaternary had an impact on the genetic structure of European temperate species in that they have undergone repeated range contractions and expansions in the wake of glacial and interglacial pulses [4]. However, the refugial populations that expanded from the south carried genetic lineages as a heritage of their long isolation from each other [5]. Nowadays, Cervidae and Odocoileinae subfamilies, representing the most common species of the Cervidae family, are in Europe. It should be mentioned that there is no available source of data concerning the actual numbers of wild animals, including Cervidae, in Europe. Although in many European countries annual numbers are monitored, the statistic is not available on the official websites of Ministries of the Environment or Statistics Offices. 

European red deer show a conspicuous phylogeographic pattern with three distinct mtDNA lineages: western, eastern, and North African/Sardinian. The western lineage, indicative of a southwestern glacial refuge in Iberia and southern France, nowadays covers large areas of the continent, including Britain, Scandinavia, and parts of central Europe, while the eastern lineage is primarily found in southeast-central Europe, the Carpathians, and the Balkans. Analyses of mitochondrial DNA expression have shown that the western lineage extends far into eastern Europe and is prominent in all eastern countries except for the Polish Carpathians, Ukraine, and Russia, where only eastern haplotypes occur. While the latter may actually reflect the natural northward expansion of the eastern lineage after the last ice age, the present distribution of the western lineage in eastern Europe may be a result of translocation and reintroduction of red deer into areas where the species became extinct in the past [6].

During the last 50 years, red deer populations and harvests have shown a general pattern of considerable increase regardless of ecological conditions, sociocultural backgrounds, or hunting systems [7]. It is widespread and abundant across much of its current range, although there is increasing fragmentation of populations in central Europe, and the species has been lost from some areas due to overhunting, agricultural intensification, and urbanization, and deliberate restriction of its range on forest management grounds. Red deer is extinct in Albania [8] and nearly extinct in Greece, where hunting is prohibited [9,10]; the few small, isolated subpopulations now existing there are the result of reintroduction into areas where it previously occurred. In Portugal, all populations have resulted from reintroduction or natural expansion from transborder Spanish populations [8].

European roe deer is the most abundant cervid occurring in Europe, with the exception of Ireland, Cyprus, Corsica, Sardinia, Sicily, and the majority of the smaller islands [11]. Historically, between the late 19th and early 20th centuries, the roe deer distribution was reduced, and their range was fragmented as a consequence of almost uncontrolled hunting and other types of human activities [12]. European roe deer population started to increase again from the beginning of the 20th century [13]. During the last half of the century, roe deer became widespread and are still expanding in many areas. Densities in the northern and southern parts of the range tend to be lower than in the central parts. During the last two decades of the 20th century, the reported roe deer numbers increased from 6.2 to 9.5 million individuals in Europe [8].

## 2. Reproductive Technology Applied to Female Deer

The application of ART in Cervidae is crucial for: (i) conservation, especially those threatened or endangered by extinction; (ii) the management of farmed deer; and (iii) understanding cellular mechanisms in the reproductive processes based on models of some species, e.g., understanding the roe deer diapause phenomenon may be useful for understanding the implantation process in other species. Diapause is a period in early neonatal development in some species during which an embryo is suspended at the blastocyst stage, and the blastocyst either expands at a very slow rate or remains totally quiescent and is thought to be obligatory in roe deer [14]. 

This manuscript is based on our published research results, the results on which we used the methodological protocols most commonly available for farm ruminants, and literature data available in the PubMed database. In vitro production of Cervidae embryos has gained prominence as a tool for use in conservation in situ and ex situ. However, the development of this technique depends on the effectivity of earlier steps that include ovarian stimulation and oocyte collection and maturation.

### 2.1. Cryopreservation of Oocytes and Embryos

Oocytes are difficult cells to cryopreserve due to their low surface area-to-volume ratio and high susceptibility to intracellular ice formation [15]. The oocyte plasma membrane differs from that of embryos, particularly due to the lack of aquaporin expression which can enhance the movement of water and cryoprotectants (CPAs). Moreover, the plasma membrane of oocytes at the second metaphase stage has a low permeability coefficient, making the movement of CPAs and water slower, and ZP acts as a barrier to the movement of water and CPAs. Oocytes have a high cytoplasmic lipid content which increases chilling sensitivity. Cryopreservation results in the generation of reactive oxygen species (ROS) which can affect viability and developmental competence [16].

At present, most investigators tend to vitrify ruminant oocytes. Vitrification represents a direct phase transition from the liquid to the glassy state, skipping the stage of ice crystal formation [17]. At present, there is no research describing cryopreservation of oocytes from cervids. Gastal et al. [18] reported the survivability of white-tailed deer ovarian tissue after cryopreservation by slow-freezing and vitrification techniques. The authors evaluated the protein expression of proliferative and apoptotic markers in ovarian tissue after thawing. Their results showed that cryopreservation of white-tailed deer ovarian tissue by either slow-freezing or vitrification did not disrupt markers of proliferation and apoptosis after thawing, and ovarian fragments cryopreserved by vitrification had greater follicle viability during in vitro culture than those that underwent slow-freezing.

Cryopreservation of deer embryos can be performed, and some details must be worked out concerning mainly the choice of cryoprotectant. Soler et al. [19] compared the effectivity of embryo transfer measured by the fertilization rate of fresh embryos with embryos briefly frozen and refrozen by vitrification and frozen with ethylene glycerol as a cryoprotectant. The authors showed that pregnancy was confirmed respectively in 18.8 and 7% of female red deer recipients, whereas embryo transfer effectivity after embryos were refrozen with glycol as a cryoprotectant resulted in pregnancies in 17–25% of deer [20,21]. 

### 2.2. Oocyte Collection

Ovarian follicles form during fetal life but there is a gap between the appearance of the first primordial follicles and the first growing primary follicles. In domestic ruminants, hyperstimulation of follicles is performed [22,23]. However, in wild animals, pharmacological stimulation is not possible to receive a greater number of follicles. Thus, in wild animals, follicles are obtained from those present on the ovary at the moment of collection. Collection of oocytes by laparoscopic ovum pick-up in sika deer was successfully implemented [24]. Nevertheless, it should be mentioned that in pubertal female red deer, the reproductive status determines the number and size of follicles present on the ovary. Our study indicates that the fourth day of the estrous cycle is the most effective period for oocyte collection (from 165 collected oocytes, 82 cumulus–oocyte complexes developed) for in vitro fertilization (IVF) and embryo development in hinds, considering the quality parameters and antioxidative potential of blastocysts [25].

For successful in vitro maturation (IVM), oocytes must undergo synchronous nuclear and cytoplasmic maturation. In most reports concerning oocytes with compact cumulus cells, they were aspirated from 3–6 mm follicles in cows, resulting in an upper limit of approximately 40% developing into good-quality blastocysts after standard maturation, fertilization, and culture procedures in vitro. If cows are treated with FSH before aspiration, the rates of development to blastocysts nearly double [26]. The aim is to create a wave of dominant follicles using FSH stimulation and continue stimulation until most of the follicles have reached the stage when LH receptors appear [27], and then mimic the rest of the natural cycle when follicles continue to grow and differentiate without FSH but with basal LH. At that time, the follicular growth pattern changes from a high mitotic period when the total granulosa and theca cell numbers increase to a point where the volume is expanded mainly by fluid accumulation with a lower mitotic index [28]. This post-FSH period is associated with an increase in oocyte quality as measured by the blastocyst rates observed after IVF [29]. In wild ruminants, hyperstimulation is excluded because making the injection is impossible and, presently, studies considering this procedure in cervids have not been described. In cervids, oocytes are collected postmortem or by laparoscopic ovum pick-up [30].

### 2.3. In Vitro Maturation (IVM)

Removal of oocytes from the inhibitory influence of the follicle allows spontaneous resumption of meiosis [31]. The oocytes mature in a special medium to the metaphase stage of meiotic second division (MII), the stage when the oocytes are released during ovulation. Bovine cumulus–oocyte complexes are matured in complex media (typically, tissue culture medium 199), supplemented with a protein source (bovine serum albumin, serum) with or without gonadotrophins (FSH, LH) and/or growth factors (e.g., epidermal growth factors). Maturation takes place in 5% CO2 in air (∼20% oxygen) in a humidified atmosphere for approximately 24 h [32]. Considering IVF in cervids, several studies have described the steps of oocyte collection, maturation, and IVF [33]. There are some differences between the protocols, but all of them are based on procedures established in cattle. The most common criterion used to evaluate the potential of oocyte and embryo development is their morphology. In the case of oocytes, it is the extent of cytoplasm compaction and formation of corona radiata cells, whereas in embryo it is color of the blastomeres, timing of blastocyst formation, and expansion and size of the embryo at hatching [34]. Recent studies have indicated some differences in gene expression associated with developmental competence in bovine and deer embryos, including the regulation of MAPK activity, translation initiation, and transcription [35]. A well-defined bovine model for in vitro investigation of oocyte developmental competence based on the time of first cleavage has been described. According to Lonergan et al. [36], good-quality embryos cleave early until 30 h post-fertilization and develop to the blastocyst stage at a greater rate than embryos with lower developmental competence, which cleave 6 h later. The differences in timing of the first cleavage result in dissimilarity in the transcriptome composition between early- and late-cleaved embryos at the two-cell stage [37,38].

Several papers have described the effectiveness of IVF measured by total number of collected oocytes used for IVM, total number of COCs used for IVF, and number of expanded blastocysts collected on particular days post-IVF. The fertilization rates reported in red deer ranged from 29.5 to 63% [25,38,39,40]. Moreover, the fertilization rate depends on the reproductive status when oocytes are collected; results of our study [25] and the Berg and Asher [41] study showed low fertilization rates during the no-breeding season. The content of the medium during IVF also plays an important role because even if successful fertilization occurs, further embryo development may be inhibited or finished before the blastocyst stage when the culture conditions are not proper, as in a study on fallow deer [42]. In the case of cervids, of interest is research connected with the differences in zona pellucida (ZP) content in selected Cervidae species that may cross naturally, e.g., Cervus elaphus and Cervus nippon [43]. Mammalian oocytes are surrounded by the ZP, a glycoprotein coat, which is considered to be an important site of sperm recognition [44,45,46]. Parillo et al. [47] observed differences in the glycosidic residue content and spatial distribution between roe, red, and fallow deer depending on the stage of follicle development.

### 2.4. Estrous Cycle Synchronization, IVF

Control of reproduction in female mammals occurs by gonadotropin-releasing hormone (GnRH), which is produced in the hypothalamus at the base of the brain and controls the release of the pituitary gonadotropins luteinizing hormone (LH) and follicle-stimulating hormone (FSH). Excitation of GnRH neurons results in the release of GnRH peptide from its secretory granules in the hypothalamus. After it diffuses into the surrounding capillary blood, GnRH travels via the hypophysial portal system to the anterior pituitary where it diffuses from the capillaries and activates LH and FSH. This activation causes the release of stored gonadotropins which diffuse back through the capillaries into the bloodstream. The gonadotropins then travel to and activate the reproductive organs, resulting in steroid synthesis and sexual activity [48].

Most Cervidae females exhibit highly seasonal patterns of autumn conception and spring parturition and are polyestrous, and readiness to fertilize (estrus phase) occurs in the autumn, with five to eight cycles expressed [49]. The average duration of the normal estrous cycle ranges from 18 to 20 days [50]. Anestrus is characterized by low peripheral plasma concentrations of progesterone (P4); it indicates complete ovulatory arrest and may persist for 4 to 6 months from spring to early autumn [51]. The concentration of plasma P4 in the anestrus female red deer is about 2–3 ng/mL [51]. The concentration of P4 during pregnancy is higher (about 7.5 ng/mL) and should remain at the same level as in the luteal phase, whereas the circulating estradiol (E2) level oscillates in about 2.7 pg/mL and should not increase in pregnancy [51]. In domestic animals, synchronization of the estrous cycle allows monitoring of the reproduction of the herd and preparation of females for appropriate biotechniques (collection of oocytes, insemination, embryo transfer). Among cervids, only a few species are recognized as domestic and are bred on farms. 

Although synchronization of estrus has been described in red, fallow, and sika deer, there are only a few papers describing both the method of pharmacological estrus synchronization and the specific time interval from eCG injection to effective insemination in cervids. So far, a P4-impregnated controlled internal drug release (CIDR) device has been used for pharmacological synchronization in red deer, inserted intravaginally for 12 days, with an intramuscular injection of 200–250 IU of equine chorionic gonadotrophin (eCG) given at the time of CIDR withdrawal. Hinds exhibited estrus between 36 and 60 h later and ovulated 24 h after the onset of estrus [20,52]. We showed that pharmacological synchronization with the use of Chronogest (sponge with 30 mg of flugestone acetate) and Folligon (pregnant mare serum gonadotrophin; 1000 IU) is more effective than CIDR and Folligon in terms of the effectiveness of the fertilization rate in hinds [53]. Moreover, according to our results, evaluating pregnancy-associated glycoproteins (PAGs) and basic ovarian steroids is useful for reproductive stage monitoring in Cervidae females. 

The red deer is a suitable model for the development of in vitro technology for endangered species. Pregnancy rates depend not only on the selection of an appropriate collection protocol and the preparation of oocytes for fertilization, but also on the collection and preparation of semen—collection method, type of semen used (frozen—thawed or fresh), quality parameters (concentration and mobility), ability to penetrate—as a key aspect after IVM [54]. The reported pregnancy rate after IVF in red deer ranged from 29.5 up to 63% [37,38,41]. Our results, measured by the number of cleaved blastocysts, indicated a rate of 36% [25]. 

## 3. Measurement of Stress

Stress is often associated with negative consequences, including decreased immune response and increased susceptibility to disease, reduced growth, and decreased reproductive performance. These adverse effects can impact individual fitness and, ultimately, population dynamics [55]. Stress is defined as anything that throws the body out of homeostatic balance. Any stressor that activates the hypothalamic–pituitary–adrenal axis (HPA) leads to increased concentrations of the adrenal stress hormone cortisol. Free-living vertebrates usually exhibit seasonal variations in cortisol levels in association with changes in activity patterns and/or environmental conditions [56]. In the past 25 years, cortisol levels in urine, hair, and feces have been increasingly used to measure stress levels in laboratory, domestic, zoo, and free-living animals [57]. The release of cortisol has an ultradian pulsatile rhythm and is influenced by a wide variety of factors and immediately increases in case of stressful events. Blood cortisol has high intra-individual variations [58]. 

Another major hypothalamic stress hormone, which is stimulated by different stressors, is vasopressin (AVP). However, measuring circulating AVP levels is challenging because it is released in a pulsatile pattern, is unstable, and is rapidly cleared from plasma. Precursor peptide (pre-provasopressin, pro-AVP) along with copeptin, which is released in an equimolar ratio to AVP, is more stable in the circulation and easy to determine [59]. The potential valuable stress marker may be copeptin. Copeptin concentration is not associated with age, lacks a consistent circadian rhythm contrary to cortisol, and increases after exercise [58].

As an example of the influence of stress on the behavior of roe deer living in two habitats in Poland, more than 60% covered by forest and more than 70% covered by agriculture fields, we measured the concentration of copeptin. The experimental material was blood plasma collected from field and the forest ecotypes of female roe deer during two hunting seasons (December/January 2019/2020 and 2020/2021).

The measurement of copeptin in plasma by EIA (EIA-COP 031921, RayBiotech, Norcross, GA, USA). The standard curve values ranged from 0.1 to 1000 ng/mL. The intra-assay and inter-assay coefficients of variation (CVs) were ˂10% and ˂15%, respectively. The copeptin concentration was higher in the blood samples of field ecotype roe deer compared to forest ecotype (*p* < 0.05; Figure 1). The results may be basic for the hypothesis that the deer living in the field biotope are more exposed to stressors related to natural environmental challenges, which means that stress can affect the survival and reproductive processes of roe deer living in a field ecotype.

Measurement of hormones used as stress markers helps understand how stress affects the survival and reproductive success of wild animals, thus how natural environmental challenges (e.g., conspecifics, predators, weather), climate change, relocation or reintroduction, and habitat disturbances impact populations [57].

## 4. Summary

Although nature itself regulates interactions within habitat abundance, the use of reproductive biotechnology in endangered populations is justified and it is possible to gain practical knowledge by mastering them with common Cervidae models. All reproductive biotechniques applied to female cervids are adopted from domestic animals and, nowadays, these techniques include synchronization of the estrous cycle, artificial insemination, superovulation, oocyte collection, IVM, IVF, embryo transfer, and cryopreservation of oocytes and embryos (Figure 2). Some reproductive biotechniques, such as IVF, have been adopted for female cervids with satisfactory results, while others, such as cryopreservation of oocytes and embryos, still require methodological refinement.

## Figures and Tables

**Figure 1 animals-11-02933-f001:**
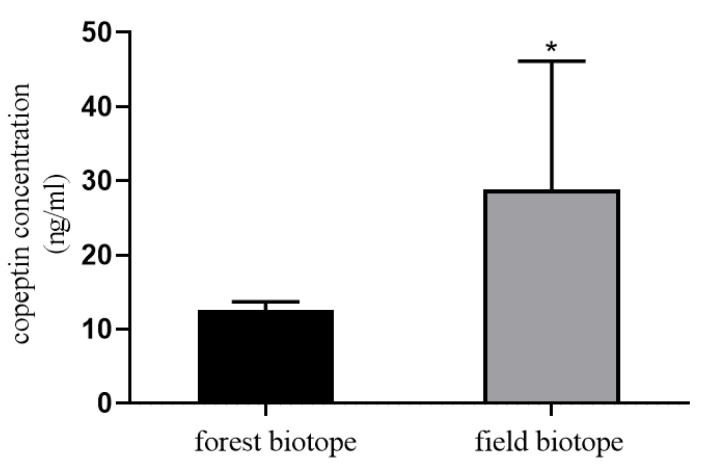
Copeptin concentration in blood plasma of roe deer from forest and field ecotype. Statistical differences were evaluated by t-test using GraphPad Prism. Asterisk (*) means statistical difference: *p* < 0.05.

**Figure 2 animals-11-02933-f002:**
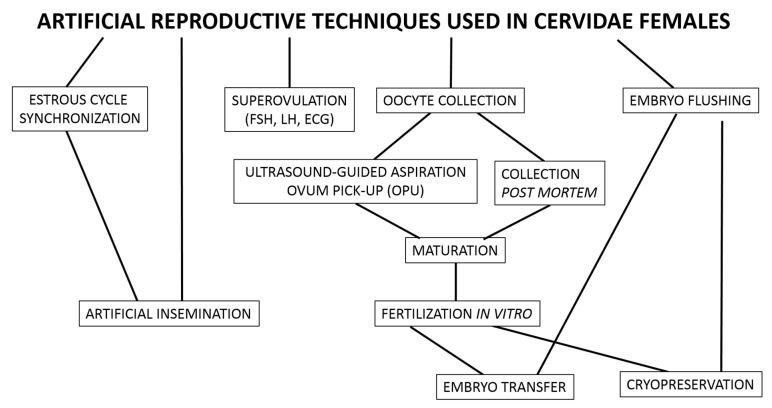
Artificial reproductive techniques used in Cervidae females.

## Data Availability

Not applicable.

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
