# Peer review of "Artificial Reproductive Technology (ART) Applied to Female Cervids Adapted from Domestic Ruminants"

_animals, 2021, doi:10.3390/ani11102933_

Round 1

Reviewer 1 Report

General comments:

The manuscript does not present enough data to support conclusions (for example: lines 278-294). No or few references, data or other evidence were used to support statements (for example 223).

There is a lot of details missing; for example: 3B, which genes, what morphological changes).

Be consistent with nomenclature, do not mix common and scientific names (ex.: line 160); common name (Genus species).

line 194: GnRH binds to GnRHR in the gonadotrophs of the anterior pituitary, not to LH/FSH

line 204-206: actual levels missing

The idea for the review is good and timely, however, I feel it lacks a common thread to tie the sections together, and the level of detail is not great enough.

Author Response

Response to Reviewer I comments

The authors are thankful for an accurate and substantive review of our manuscript. All comments and suggestions are valuable for us and we thought about Your suggestions.

The manuscript does not present enough data to support conclusions (for example: lines 278-294).

Answer: We agree that copeptin concentration measurement is a novel. Because of that we changed the conclusion and changed “this indicates…” for  „the results may be basic for the hypothesis that…”

No or few references, data or other evidence were used to support statements (for example 223).

Answer: We tried at first to review the available knowledge based on ART in Cervids but as we mentioned, relatively small data describe the techniques in these ruminants. Many techniques and protocols are adopted from cows and sheeps for Cervids. We agree that in domestic animals, ART are described very well and are practiced. Nevertheless the aim of this review was focus on ART already used on Cervids only and the references, in our opinion should be limited to Cervids.

There is a lot of details missing; for example: 3B, which genes, what morphological changes). Be consistent with nomenclature, do not mix common and scientific names (ex.: line 160); common name (Genus species).

Answer: We corrected the names of genius thought the manuscript and precised the selected genes according to Your suggestion.

line 194: GnRH binds to GnRHR in the gonadotrophs of the anterior pituitary, not to LH/FSH.

Answer:  We apologize for mistake, we proved it.

line 204-206: actual levels missing

Answer: We added the concentration level.

The idea for the review is good and timely, however, I feel it lacks a common thread to tie the sections together, and the level of detail is not great enough.

Answer: We tried to prove the manuscript according to all review comments and in our opinion, more details are added in the proved form which satisfies the Reviewer.

Reviewer 2 Report

Altogether this is an interesting and well-written review. A systematic review, however, might have been a better approach. At present it is unclear if all relevant references with regard to this topic were adequately considered. The authors should include some information how they searched for literature into the manuscript.

The relevance of Figure 1 is questionable.

Figure 2 could be improved, it is very general,

Author Response

The authors are thankful for an accurate and substantive review of our manuscript. All comments and suggestions are valuable for us and we thought about Your suggestions.

Altogether this is an interesting and well-written review. A systematic review, however, might have been a better approach. At present it is unclear if all relevant references with regard to this topic were adequately considered. The authors should include some information how they searched for literature into the manuscript.

Answer: We added the explanation in the manuscript.

The relevance of Figure 1 is questionable.

Answer: Copeptin is a novel sress marker comparing with cortisol. For us it is very important to point for convenience of copeptin, which is stable factor and without daily fluctuations. Thus this preliminary results presented in Figure 1 are very important for us to publish. Of course we are planning the validation of copeptin in other samples. Because of that we changed the conclusion and changed “this indicates…” for  „the results may be basic for the hypothesis that…”

Figure 2 could be improved, it is very general

Answer: We agree that the schema is general but adding more details in our opinion make this schema unreadable. This schema aim is summary the ART in Cervids.

Reviewer 3 Report

The manuscript "Artificial reproductive technology (ART) applied to female cervids adapted from domestic ruminants" is a review article summarizing the current status of ART in deer.

In general, the manuscript ranges from very superficial information (leaving out important facts) to very detailed explanations that may not be relevant. Additionally, this review finds some sessions in the article disorganized and suggests that they get reevaluated.

Specific comments follow:

  • line 87. Subtitle 2 with only 5 lines is unnecessary. This information can be part of the current heading 3. 
  • line 91. The diapause phenomenon is mentioned here and nowhere else in the paper, there is not enough context to its mention.
  • line 98. This section needs to be edited and reorganized, there is a repetition of statements, there are mistakes (in line 101 hyperstimulated follicles, not oocytes), additional review of the literature is needed (line 124 indicates that hyperstimulation for OPU has not been described in wild deers and it has). There is no information on the literature review of the numbers of oocytes collected postmortem or per opu.
  • line 152 as you are reporting a range from different  references I suggest you keep it as a range (...in red deer ranged from 29.5% to 63% [16, 29-31] )
  • line 186 This section contains a good summary of the reproductive physiology of deer and some more basic ART like synchronization and AI. I strongly suggest that this becomes the first section in the reproductive technology subheading ( before diving into oocyte collection etc)
  • line 205 for completion, it will be good to note the progesterone value on pregnant deer.
  • line 219 these are commercial products, you should write also/or the active ingredient ( ecg and flugestone acetate respectively )
  • 223 this paragraph on IVF should be integrated into the paragraph in line 149 to make the paper easier to read and in part, because there are lines that are exactly the same ( line 151 and 228 )
  • 231 this paragraph is out of context and with mistakes. 1st. This talks about embryo flushing (a topic that has not been mentioned so far in this paper) 2. it says embryo flushing of the recipient. recipients are not flushed, are transfer, so you may want to say, donor. I suggest extending the information on MOET or leaving it completely out.
  • line 186 Also, The title of the session said "embryo transfer" yet, the topic is not mentioned at all. either remove it from the title or talk about embryo transfer whether from MOET or IVF and include pregnancy rates 
  • 235 this paragraph has to be rewritten. the term "fertilization rate", the effectivity of embryo transfer"  and 'embryo transfer effectivity" should be changed to pregnancy rate. but in general, the paragraph needs to be rewritten, is confusing, the references are missed placed and it does not use appropriate terminology. Also since most of the paper is focused on IVF and conventional embryo collection has not been mentioned, it will be important to state the current status of embryo cryopreservation for IVF embryos and clarify that in this paragraph you are referring to MOET embryos. 
  • line 242 Glycerol not glycol
  • line 244 not sure how all of this fits in the ART manuscript, but regardless the title is misleading.... you are not talking about the influence of stress on animal welfare you are talking about methods to measure stress. In general, there is too much information related to the measurement of stress and yet not citations correlating stress markers and reproduction. 
  • line 278 the summary of this study needs to be rewritten and reviewed by a native English speaker.  Also, you refer to a study that has not been published (peer-reviewed) and that needs to be stated. Also,  "higher copeptin group are more exposed to stress that can affect survival and reproductive processes " can not be concluded. so far you just have a population with a higher copeptin and that will need to be correlated with survival and reproduction in order to make such a conclusion. maybe leave it as a hypothesis 

figure 1. may not be relevant for the paper. 

Schema. is good but don't recall seeing it referenced during the manuscript. Also, consider embryo flushing as an event after superovulation

Author Response

The authors are thankful for an accurate and substantive review of our manuscript. All comments and suggestions are valuable for us and we thought about Your suggestions.

The manuscript "Artificial reproductive technology (ART) applied to female cervids adapted from domestic ruminants" is a review article summarizing the current status of ART in deer.

In general, the manuscript ranges from very superficial information (leaving out important facts) to very detailed explanations that may not be relevant. Additionally, this review finds some sessions in the article disorganized and suggests that they get reevaluated.

Specific comments follow:

line 87. Subtitle 2 with only 5 lines is unnecessary. This information can be part of the current heading 3. 

Answer: We changed it according to Your suggestion.

line 91. The diapause phenomenon is mentioned here and nowhere else in the paper, there is not enough context to its mention.

Answer: We added several sentences concerning diapause according to Your suggestion.

line 98. This section needs to be edited and reorganized, there is a repetition of statements, there are mistakes (in line 101 hyperstimulated follicles, not oocytes), additional review of the literature is needed (line 124 indicates that hyperstimulation for OPU has not been described in wild deers and it has). There is no information on the literature review of the numbers of oocytes collected postmortem or per opu.

Answer: We added several sentences concerning diapause according to Your suggestion based on paper of Lambert et al. and the indicated section is proved according to suggestions.

line 152 as you are reporting a range from different  references I suggest you keep it as a range (...in red deer ranged from 29.5% to 63% [16, 29-31] )

Answer: We changed it according to Your suggestion.

line 186 This section contains a good summary of the reproductive physiology of deer and some more basic ART like synchronization and AI. I strongly suggest that this becomes the first section in the reproductive technology subheading ( before diving into oocyte collection etc)

Answer: We changed it according to Your suggestion.

line 205 for completion, it will be good to note the progesterone value on pregnant deer.

Answer: We added the information about progesterone concentration.

line 219 these are commercial products, you should write also/or the active ingredient ( ecg and flugestone acetate respectively )

Answer: We added the information about active substances present in products.

223 this paragraph on IVF should be integrated into the paragraph in line 149 to make the paper easier to read and in part, because there are lines that are exactly the same ( line 151 and 228 )

Answer: We joined the paragraphs.

231 this paragraph is out of context and with mistakes. 1st. This talks about embryo flushing (a topic that has not been mentioned so far in this paper) 2. it says embryo flushing of the recipient. recipients are not flushed, are transfer, so you may want to say, donor. I suggest extending the information on MOET or leaving it completely out.

Answer: We deleted the paragraph.

line 186 Also, The title of the session said "embryo transfer" yet, the topic is not mentioned at all. either remove it from the title or talk about embryo transfer whether from MOET or IVF and include pregnancy rates.

Answer: We deleted embryo transfer from the title.

235 this paragraph has to be rewritten. the term "fertilization rate", the effectivity of embryo transfer"  and 'embryo transfer effectivity" should be changed to pregnancy rate. but in general, the paragraph needs to be rewritten, is confusing, the references are missed placed and it does not use appropriate terminology. Also since most of the paper is focused on IVF and conventional embryo collection has not been mentioned, it will be important to state the current status of embryo cryopreservation for IVF embryos and clarify that in this paragraph you are referring to MOET embryos. We changed fertilization rate for pregnancy rate.

Answer: We collected oocytes not by MOET but by aspiration post mortem than we can`t agree for Your further suggestions.

line 242 Glycerol not glycol

Answer: We changed it according to Your suggestion.

line 244 not sure how all of this fits in the ART manuscript, but regardless the title is misleading.... you are not talking about the influence of stress on animal welfare you are talking about methods to measure stress. In general, there is too much information related to the measurement of stress and yet not citations correlating stress markers and reproduction. 

Answer: We changed and shortened the paragraph.

line 278 the summary of this study needs to be rewritten and reviewed by a native English speaker.  Also, you refer to a study that has not been published (peer-reviewed) and that needs to be stated. Also,  "higher copeptin group are more exposed to stress that can affect survival and reproductive processes " can not be concluded. so far you just have a population with a higher copeptin and that will need to be correlated with survival and reproduction in order to make such a conclusion. maybe leave it as a hypothesis 

figure 1. may not be relevant for the paper.

Answer: We are keen to leave the results on the concentration of copeptin, because it is promising and helpful protein. unlike the measurement of cortisol levels, there are no daily fluctuations in the concentration of copeptin. Of course we are planning the validation of copeptin in other samples. Because of that we changed the conclusion and changed “this indicates…” for  „the results may be basic for the hypothesis that…”

Schema. is good but don't recall seeing it referenced during the manuscript. Also, consider embryo flushing as an event after superovulation.

 Answer: According to the published results superovulation of hinds during pregnancy is not practiced thus in our opinion the addition of this biotechnique is actually prematurely. Such ART is practiced in domestic animals.